# Measurement report: Molecular composition, sources, and evolution of atmospheric organic aerosols in a basin city in China

Junke Zhang<sup>1</sup>, Xinyi Fu<sup>1</sup>, Chunying Chen<sup>1</sup>, Yunfei Su<sup>1</sup>, Siyu Liu<sup>1</sup>, Luyao Chen<sup>2</sup>, Yubao Chen<sup>2</sup>, Gehui Wang<sup>2</sup>, Andre S. H. Prevot<sup>3</sup>

- <sup>1</sup>School of Environmental Science and Engineering, Southwest Jiaotong University, Chengdu 611756, China
  - <sup>2</sup>Key Lab of Geographic Information Science of the Ministry of Education, School of Geographic Sciences, East China Normal University, Shanghai 200241, China
  - <sup>3</sup>PSI Center for Energy and Environmental Sciences, Paul Scherrer Institute, Villigen 5232, Switzerland

Correspondence to: Junke Zhang (<u>zhangjunke@home.swjtu.edu.cn</u>), Gehui Wang (<u>ghwang@geo.ecnu.edu.cn</u>), and Andre S.

H. Prevot (andre.prevot@psi.ch)

Abstract. Although organic aerosols (OA) have important impacts on climate, environment, and health, research on OA in Sichuan Basin (SCB), one of the heavily polluted areas in China, is still scarce. In this study, the PM<sub>2.5</sub> samples were collected during winter 2023 in Chengdu, the capital city of Sichuan Province, and analyzed for organic compounds using gas chromatography-mass spectrometry. The total average concentration of 125 organic compounds was  $2013.4 \pm 902.4 \text{ ng m}^{-3}$ (mean ± standard deviation), and they were dominated by fatty acids (28.9%), phthalate esters (28.4%), and anhydrosugars (18.0%). Anthropogenic sources such as fossil fuel and biomass burning were the main sources of aliphatic lipids. Softwood burning was the main source of anhydrosugars. Although both are related to the aging of polycyclic aromatic hydrocarbons (PAHs), oxygenated PAHs and phthalic acids demonstrated different generation mechanisms. The isoprene secondary OA (SOA) tracers were strongly affected by NOx, relative humidity, and aerosol acidity. Biomass burning was an important source of biogenic SOA tracers. Tracer-based methods revealed that anthropogenic sources (11.6%), β-caryophyllene (11.0%), and biomass burning (10.0%) were important sources of organic carbon (OC). Positive matrix factorization (PMF) analysis demonstrated that secondary formation (22.2%) was the greatest source of OC, followed by dust (20.4%), vehicular emissions (17.6%), plastic related sources (17.4%), biomass burning (11.3%), coal combustion (6.2%), and primary biogenic emissions (5.0%). As pollution worsened, the proportions of secondary inorganic species and secondary OC in PM<sub>2.5</sub> increased substantially; PMF analyses indicated that OC increase was caused mainly by secondary formation and biomass burning. These results are of great value for understanding the characteristics and formation mechanisms of OA, and its contribution to air pollution in the SCB.

#### 1 Introduction





As the world's largest developing country, China has experienced rapid economic growth and substantial urbanization development over recent decades. This has resulted in large amounts of pollutants being discharged into the atmosphere in association with coal and biofuel burning, vehicular emissions, industrial activities, dust, and waste incineration. China is still

a hot spot for air pollutant emissions albeit significant pollution reductions in the past decade. Accordingly, haze events caused by particulate matter with aerodynamic diameter of  $\leq 2.5 \,\mu m$  (PM<sub>2.5</sub>), the most important atmospheric pollutant, have occurred frequently in some regions of China, e.g., the North China Plain, Yangtze River Delta, Pearl River Delta, Fen Wei Plain, and Sichuan Basin (SCB) (Zhang et al., 2014; Wang et al., 2014; Yan et al., 2024; Huang et al., 2018; Huang et al., 2014; Zhang et al., 2024e; Wang et al., 2016; Wu et al., 2020; Ren et al., 2024; Ren et al., 2016; Wang et al., 2025). Organic aerosols (OA), which comprise thousands of organic compounds, contribute approximately 20%-50% of the total mass of PM<sub>2.5</sub> in the continental mid-latitudinal atmosphere, whereas the contribution is approximately 90% in areas of tropical forest (Cui et al., 2023; Chen et al., 2022; Deshmukh et al., 2019). In China, OA is considered an important contributor to heavy pollution. For example, during extremely severe haze pollution events that occurred in January 2013 in China, OA constituted the major fraction (30%-50%) of PM<sub>2.5</sub>, and secondary OA (SOA) contributed up to 70% of the OA (Huang et al., 2014). Previous studies demonstrated that OA can affect Earth's climate by warming the atmosphere (absorbing solar radiation), cooling the atmosphere (scattering solar radiation), and altering the glacier/snow albedo in the cryosphere system (Chen et al., 2022; Dong et al., 2024; Wan et al., 2017). Moreover, various toxic substances found in OA, such as polycyclic aromatic hydrocarbons (PAHs), can enter the food chain and pose serious risk to the health of the entire natural ecosystem (Yang et al., 2018; Wang et al., 2009). Therefore, the in-depth research on OA is of great value for understanding its climate effects and air pollution control.







Atmospheric OA, derived from anthropogenic and natural sources, can be divided into primary OA (POA) and SOA. The former is derived directly from sources such as fossil fuel combustion, industrial activities, biomass burning, soil dust, cooking emissions, and primary biogenic emissions; the latter, produced by the reactions of volatile organic compounds (VOCs) with ozone (O<sub>3</sub>), hydroxyl (OH), and nitrate (NO<sub>3</sub>) radicals, is formed through nucleation, condensation on and/or uptake by preexisting particles (Deshmukh et al., 2019; Chen et al., 2022). Compared with POA, most SOA is usually distributed in a smaller particle size range, which makes them more prone to accumulating toxic and harmful substances, and affords them longer retention time in the atmosphere. Additionally, SOA typically has greater polarity, oxidative potential, hygroscopicity, and solubility. All these attributes mean that SOA generally has more profound or serious negative impacts than those associated with POA (Wang et al., 2009; Wang et al., 2006; Fu et al., 2011; Daellenbach et al., 2020).

The chemical composition of OA is extremely complex. Although various analytical techniques have been used to analyze the OA components in different environmental media, only 10%–30% of organic compounds of ambient particulate matter mass can be identified (Noziere et al., 2015). Determination of organic components, especially biomarkers that can indicate the source and generation mechanism of OA, is of great practical importance for studying the source, formation, and transformation of OA (Deshmukh et al., 2019). For example, phthalate esters have been used widely as plasticizers and

softeners in the production processes of resins and plastics (Fu et al., 2008; Fu et al., 2010), PAHs are often used as organic biomarkers for coal combustion emissions (Fu et al., 2008), levoglucosan is used widely as a biomarker for biomass burning (Wang et al., 2021; Deshmukh et al., 2019), and hopanes are specific biomarkers of petroleum and coal (Fu et al., 2008). Additionally, certain characteristic parameters or ratios of molecular markers can also be used to analyze their sources. For example, the levoglucosan/mannosan ratio is a robust indicator for distinguishing combustion materials (e.g., softwood, hardwood, herbs, and straw) (Deshmukh et al., 2019; Wang et al., 2021).







The SCB, located in southwestern China, is one of the most densely populated areas in China, and the level of emission of atmospheric pollutants in this region is high. Geographically, the SCB is surrounded by mountains and plateaus with elevation of 2000-3000 m, which often cause light winds and stagnant weather conditions within the basin. This situation hinders dispersion of pollutants emitted locally, ultimately making the SCB one of the regions of China with the most severe air pollution. Chengdu, the capital of Sichuan Province, had an average annual PM<sub>2.5</sub> concentration of 39 μg m<sup>-3</sup> in 2023, which is higher than that observed in the representative cities of other polluted areas in China, e.g., Beijing (32 µg m<sup>-3</sup>), Shanghai (28 μg m<sup>-3</sup>), Guangzhou (23 μg m<sup>-3</sup>), and Nanjing (29 μg m<sup>-3</sup>) (https://www.cnemc.cn/). Under the influence of the regional topography and climate, the characteristics and formation mechanisms of pollution in the SCB also differ markedly from those of other regions (Huang et al., 2018; Huang et al., 2021; Zhang et al., 2024e; Wang et al., 2018a; Zhang et al., 2024b). Many previous studies found that OA represents an important chemical component of the PM<sub>2.5</sub> in the SCB (Zhang et al., 2024d; Zhang et al., 2024c; Wang et al., 2018a; Zhang et al., 2023). We also found that the contribution of OA in winter in Chengdu has increased annually over recent years, and it is expected that this contribution will increase further with ongoing aggravation of regional pollution (Zhang et al., 2024a). However, most previous studies on OA in this region treated them as a group, analyzing its overall concentration or contribution to PM<sub>2.5</sub>, or evaluating the relative proportions of primary and secondary organic carbon (OC) (Zhang et al., 2024e; Zhang et al., 2024c; Tao et al., 2014). Research on determination and analysis of OA at the molecular level remains very limited, and only a few studies have investigated certain types of OA species (Cui et al., 2023; Li et al., 2019b; Tao et al., 2014; Zhao et al., 2020; Yang et al., 2018), hindering comprehensive understanding of the atmospheric OA in the SCB. To achieve more comprehensive understanding of atmospheric OA in the SCB, we collected day/night PM<sub>2.5</sub> samples in Chengdu during winter 2023, and quantitatively analyzed the concentrations of 15 compound classes (125 organic species). Based on the results, the concentration levels, molecular composition, sources, and evolution of the derived organic compounds were investigated. To the best of our knowledge, this work represents one of the most comprehensive studies of ambient OA in the SCB at the molecular level. The findings are of great importance for improved understanding both of the characteristics of OA in the SCB and of their regional climatic and environmental impacts.

#### 2 Experimental methods

#### 2.1 Site and sampling




Chengdu, the capital of Sichuan Province, had a permanent population of 21.4 million in 2023 (https://cdstats.chengdu.gov.cn/). It is also the city in China with the highest number of motor vehicles, i.e., in excess of 6 million vehicles in 2023 (https://www.mee.gov.cn/). In this study, daytime and nighttime PM<sub>2.5</sub> samples were collected at the Jiuli Campus of Southwest Jiaotong University, located in the center of Chengdu city, throughout the period extending from 14 December 2023 to 15 January 2024. The study site is in an area of mixed educational, commercial, transportation, and residential land uses (Fig. SI). The lack of obvious local pollution sources means that this site is representative of a typical urban atmospheric environment affected by pollution from multiple sources.

The sampler (Tisch Environmental, USA) was placed on the rooftop of a seven-story building (approximately 25 m above ground level) without a shelter, and it operated with a sampling flow rate of 1.05 m³ min⁻¹. All samples were collected on a prebaked (450 °C for 8 h) quartz microfiber filter (210 mm×297 mm). Each filter was exposed for a period of approximately 11.5 h (daytime: 08:00–19:30, nighttime: 20:00–07:30 (local time)). After sampling, each filter was sealed in an aluminum bag and stored at −18 °C until required for analysis. Field blanks were treated using the same sampling procedure as that used for the filters, except that they were placed on the sampler for only 5 min and without turning on the pump. Ultimately, 33 pairs of daytime/nighttime samples and 4 blank samples were used in this study.

#### 2.2 Sample extraction, derivatization, and GC/MS quantification

A filter aliquot was extracted using dichloromethane and methanol (2:1, v/v) under ultrasonication (10 min each, repeated three times). The extracts were filtered through quartz fiber wool packed within a Pasteur pipette, concentrated using a rotary evaporator under vacuum conditions, and then dried using pure nitrogen. After reaction with a mixture of N,O-bis-(trimethylsilyl) trifluoroacetamide and pyridine (5:1, v/v) at 70 °C for 3 h, the derivatives were diluted using an internal standard (n-alkane  $C_{13}$ ) at a concentration of 3.024 ng  $\mu$ L<sup>-1</sup>.

Gas chromatography–mass spectrometry (GC/MS) analysis of the derivatized fraction was performed using an Agilent 7890B gas chromatograph coupled with an Agilent 5977B MSD (Agilent Company, USA). The GC separation was undertaken using an HP-5MS fused silica capillary column. The sample was injected in a splitless mode at an injector temperature of 280 °C, and scanned from 50–650 Daltons using an electron impact mode at 70 eV. All detected compounds were quantified using the peak area of the individual characteristic ion. No notable contamination (<10% of those in the samples) was found in the blanks. Average recoveries of the target compounds were better than 70%. Further details of the methods adopted for extraction, derivatization, and GC/MS analysis can be found in S1 and our previous studies (Wang and Kawamura, 2005; Wang et al.,

#### 2.3 Measurements of OC, EC, and inorganic ions

Measurements of OC and elemental carbon (EC) were performed on a punch (0.526 cm<sup>2</sup>) of each quartz sample using a thermal/optical carbon analyzer (DRI model 2001, Desert Research Institute, USA), adopting the Interagency Monitoring of Protected Visual Environments thermal evolution (IMPROVE) protocol with reflectance charring correction (Chow et al., 2007). The major cations (i.e., Ca<sup>2+</sup>, Na<sup>+</sup>, K<sup>+</sup>, Mg<sup>2+</sup>, and NH<sub>4</sub><sup>+</sup>) and anions (i.e., F<sup>-</sup>, Cl<sup>-</sup>, SO<sub>4</sub><sup>2-</sup>, and NO<sub>3</sub><sup>-</sup>) were detected using an ion chromatograph (IC) system (ICS-90, Dionex, USA).

## 2.4 PMF analysis






Positive matrix factorization (PMF) is a mathematical multivariate factor analysis method used widely with environmental sample data to identify the factors or the sources affecting the sample data and to quantify their contributions, assuming mass conservation between emission sources and receptors (Paatero and Tapper, 1994). It is a method proven suitable for source apportionment of gaseous and particulate pollutants in the atmosphere (Zhang et al., 2024e; Xu et al., 2018; Liu et al., 2016; Sun et al., 2016; Lei et al., 2023; Wu et al., 2016; Huang et al., 2017; Wan et al., 2023; Cui et al., 2023; Canonaco et al., 2021). In this study, we performed PMF analysis to investigate the sources of OC. An introduction to the principles of PMF and a description of the settings used in this study can be found in S2.

#### 3 Results and discussion

# 3.1 Pollution characteristics in winter in Chengdu

During the study period, the weather in Chengdu was generally cold and humid. The ambient temperature and relative humidity varied in the range -2.4 to 17.6 °C ( $7.2 \pm 3.9$  °C, mean  $\pm$  standard deviation) and 28%-100% ( $78.2\%\pm18.5\%$ ), respectively (Fig. 1a). No rain was recorded. The PM<sub>2.5</sub> during the study period exhibited a mass concentration of 38.6-163.7 µg m<sup>-3</sup>, with an average value of  $106.5 \pm 29.1$  µg m<sup>-3</sup>, which is 3.0 and 21.3 times greater than the Chinese National Ambient Air Quality Standard (annual average: 35 µg m<sup>-3</sup>) and the World Health Organization guideline value (annual average: 5 µg m<sup>-3</sup>), respectively. According to the Air Quality Index (AQI) statistical results, during the whole observation period (33 days in total), the number of pollution days (AQI value of >100) in Chengdu reached 20, which is higher than that observed during the same period in other polluted cities in China, e.g., Beijing (3 days), Shanghai (12 days), Guangzhou (1 day), Nanjing (11 days), and Xi'an (18 days) (https://www.aqistudy.cn/). This means that although air quality in China has improved substantially in recent years, PM<sub>2.5</sub> pollution in the SCB remains a serious problem, especially in winter.

Figure 1: Time series of (a) temperature (T) and relative humidity (RH), (b) concentrations of PM<sub>2.5</sub>, OM, EC, and inorganic ions, and (c) concentrations of OA compounds (SOA<sub>1</sub>: isoprene SOA tracers, SOA<sub>M</sub>: monoterpene SOA tracers, SOA<sub>S</sub>: sesquiterpene SOA tracers).




The average mass concentrations of organic matter (OM, =OC×1.6) and EC were  $26.6 \pm 10.1$  and  $5.5 \pm 1.5 \,\mu g \, m^{-3}$ , accounting for 24.7% and 5.3% of the PM<sub>2.5</sub> mass, respectively (Fig. 1b). The contribution of OM is comparable with results obtained in previous winters in Chengdu, e.g., 22.2% in 2011 (Tao et al., 2014) and 27.8% in 2014 (Wang et al., 2018a). The average total concentration of water-soluble inorganic ions was  $47.1 \pm 20.7 \,\mu g \, m^{-3}$ , which explained 42.5% of the PM<sub>2.5</sub> mass. Among the water-soluble inorganic ions, NO<sub>3</sub><sup>-</sup> ranked the highest (25.1  $\pm$  12.8  $\mu g \, m^{-3}$ ), followed by NH<sub>4</sub>\* (8.4  $\pm$  4.2  $\mu g \, m^{-3}$ ) and SO<sub>4</sub><sup>2</sup>- (7.7  $\pm$  3.0  $\mu g \, m^{-3}$ ), which accounted for 22.2%, 7.5%, and 7.1% of PM<sub>2.5</sub> mass, respectively. Compared with the concentrations and contributions of NO<sub>3</sub><sup>-</sup> (15.5  $\pm$  5.4  $\mu g \, m^{-3}$ , 9.8%), NH<sub>4</sub>\* (15.3 $\pm$ 5.7  $\mu g \, m^{-3}$ , 9.7%), and SO<sub>4</sub><sup>2</sup>- (31.8 $\pm$ 10.7  $\mu g \, m^{-3}$ , 20.1%) in winter 2011 in Chengdu (Tao et al., 2014), the concentration and contribution of NO<sub>3</sub><sup>-</sup> increased by 9.6  $\mu g \, m^{-3}$  and 12.4%, respectively, while the concentration and contribution of NH<sub>4</sub>\* decreased by 6.9  $\mu g \, m^{-3}$  and 2.2%, respectively. Meanwhile, the concentration and contribution of SO<sub>4</sub><sup>2-</sup> experienced the largest decrease, with a decrease of 24.1  $\mu g \, m^{-3}$  and 13.0%, respectively. This is directly related to the different emission reductions from precursors of these inorganic species (e.g., SO<sub>2</sub>, NOx, and NH<sub>3</sub>) in recent years. For example, the annual average SO<sub>2</sub> and NO<sub>2</sub> concentrations in Chengdu have decreased by 84.2% and 59.3% respectively from 2014 to 2024 (https://sthj.chengdu.gov.cn). The same phenomenon has also been observed in other areas in China, such as Beijing-Tianjin-Hebei (Li et al., 2021a) and Yangtze River Delta (Shen et al., 2020). In addition,

the  $NO_3^-/SO_4^{2-}$  mass ratio reached  $3.1 \pm 0.9$ , which is higher than that observed in previous studies in winter in Chengdu, e.g., 1.1 in 2014 (Wang et al., 2018a) and 2.4 in 2022 (Zhang et al., 2024e). This annual increase in the ratio emphasizes the increasingly important role played by mobile sources (mainly motor vehicles) in relation to air pollution in Chengdu.

#### 3.2 Molecular composition of OA







In this study, 125 organic compounds were detected in the PM<sub>2.5</sub> samples, which were grouped into 15 classes. The range and mean values of the concentrations of all organic compounds are listed in Table S1. Figure 1c shows the chemical compositions of the organic compounds found in individual samples. The concentrations of all the quantified organic compounds in Chengdu was  $2013.4 \pm 902.4$  ng m<sup>-3</sup>, with predominance of fatty acids ( $582.0 \pm 458.4$  ng m<sup>-3</sup>), phthalate esters ( $571.8 \pm 246.7$  ng m<sup>-3</sup>), and anhydrosugars ( $362.3 \pm 162.1$  ng m<sup>-3</sup>), which contributed 28.9%, 28.4%, and 18.0% to the total determination of organic compounds, respectively. However, certain organic compounds were present at very low concentrations, i.e., nitrophenol, isoprene SOA tracers (SOA<sub>1</sub>), PAHs, sugar alcohols, and hopanes, and the contribution of each was <1%.

# 3.2.1 Aliphatic lipids

**n-Alkanes.** In this study, the range of the concentration of n-alkanes ( $C_{18}$ – $C_{36}$ ) was 49.0–286.8 ng m<sup>-3</sup>, with the average of 139.9 ± 60.0 ng m<sup>-3</sup>, which is in reasonable agreement with results reported for winter in Chennai (141 ± 110 ng m<sup>-3</sup>) in India (Fu et al., 2010), but lower than the results reported for winter in Greater Cairo (653.8 ± 641.2 ng m<sup>-3</sup>) in Egypt (Farah et al., 2024) and in Tianjin (343 ± 227 ng m<sup>-3</sup> during daytime, 499 ± 307 ng m<sup>-3</sup> at nighttime) in North China (Fan et al., 2020). Additionally, the concentration of n-alkanes in Chengdu is much higher than those over central Alaska (24 ± 23 ng m<sup>-3</sup>) (Deshmukh et al., 2019) and those in marine aerosols over the Arctic Ocean (0.14–4.5 ng m<sup>-3</sup>) (Fu et al., 2013).

As shown in Fig. S2, odd carbon number dominance was found for the higher molecular weight (HMW; >C26) n-alkanes, with carbon chain lengths of C<sub>29</sub> and C<sub>31</sub> being dominant, whereas lower molecular weight (LMW; ≤C26) n-alkanes showed weaker odd/even predominance. This is consistent with results reported for winter in Beijing (Yang et al., 2023) and Nanchang (Guo et al., 2024). According to previous studies, HMW n-alkanes reflect sources such as biomass burning and waxes in terrestrial plants, while LMW n-alkanes are derived mainly from anthropogenic sources such as fossil fuel combustion and vehicular emission (Fu et al., 2010; Yamamoto and Kawamura, 2010). We found that the concentration of HMW n-alkanes was 2.8 times greater than that of LMW n-alkanes, which is comparable with results obtained in Nanchang (HMW/LMW = 3.22) (Guo et al., 2024), but contrary to observations in Beijing, where the concentration of LMW n-alkanes was 1.2 times greater than that of HMW n-alkanes (Yang et al., 2023). This might reflect greater contribution from coal-fired sources in winter in North China, whereas there is greater contribution from biomass burning in winter in the SCB, as confirmed by the strong positive correlation found between n-alkanes and levoglucosan (R² = 0.74) in our study.

A value of the carbon preference index (CPI) close to 10 is considered to reflect plant wax emission, whereas a value near unity indicates emissions derived mainly from anthropogenic activities, e.g., fossil fuel combustion (Simoneit et al., 1991; Rogge et al., 1993). The calculated CPI of the n-alkanes in Chengdu was  $1.74 \pm 0.25$ . Therefore, we can conclude that the n-alkanes were derived mainly from anthropogenic sources rather than from higher plant wax emissions. It is worth noting that the value of the CPI obtained in our study is slightly higher than that reported for winter in parts of northern China, e.g., Tianjin  $(1.21 \pm 0.11)$  during daytime,  $1.19 \pm 0.09$  at nighttime) (Fan et al., 2020) and Beijing (1.1) (Wang et al., 2020). This is mainly because the prevalence of coal-fired heating in winter in northern cities of China results in a value of the CPI that was closer to 1. Additionally, the CPI value in Chengdu is much lower than that reported in mountainous areas such as Mt. Tai in China (4.42 during daytime, 4.63 at nighttime) (Fu et al., 2008), and in Alaska in the USA (6.6) (Deshmukh et al., 2019).

Fatty acids. A homologous series of straight-chain fatty acids ( $C_{10:0}$ – $C_{32:0}$ ), including unsaturated ( $C_{18:1}$ ) acids, was detected in the Chengdu aerosols. The average concentration of the fatty acids was  $582.0 \pm 458.4$  ng m<sup>-3</sup>, with a higher concentration at nighttime ( $726.3 \pm 560.6$  ng m<sup>-3</sup>) than during daytime ( $437.6 \pm 263.8$  ng m<sup>-3</sup>). This concentration of fatty acids is lower than that reported for winter in Tianjin ( $666 \pm 418$  ng m<sup>-3</sup> during daytime,  $778 \pm 448$  ng m<sup>-3</sup> at nighttime) (Fan et al., 2020), but higher than that observed in Chennai ( $302 \pm 179$  ng m<sup>-3</sup>) (Fu et al., 2010) and Beijing (207.6 ng m<sup>-3</sup>) (Wang et al., 2020).

Previous studies identified that HMW ( $C_{20:0}$ – $C_{34:0}$ ) fatty acids are derived from terrestrial higher plant wax, whereas LMW ( $< C_{20:0}$ ) fatty acids have multiple sources such as vascular plants, microbes, marine phytoplankton, kitchen emissions, and biomass burning (Rogge et al., 1991; Schauer et al., 2001). In this study, the LMW/HMW fatty acids concentration ratio was 1.63 during daytime and 2.73 at nighttime, i.e., lower than the ratio of  $5.3 \pm 1.8$  in summer but higher than the ratio of  $1.4 \pm 0.8$  in winter in Chennai (Fu et al., 2010). Meanwhile, the day–night difference in concentration is different from that based on observations in winter in northern China, e.g., the higher ratio during daytime (3.35) than at nighttime (2.77) in Tianjin (Fan et al., 2020). This might be attributed to greater biomass burning and to a lower boundary layer height during the night in winter in Chengdu.

Consistent with some megacities in China, such as Tianjin (Fan et al., 2020) and Beijing (Wang et al., 2020), the distributions of fatty acids in this study were found characterized by a strong even carbon number predominance with maxima at  $C_{16:0}$  (137.0  $\pm$  83.3 ng m<sup>-3</sup>) and  $C_{18:0}$  (90.2  $\pm$  98.3 ng m<sup>-3</sup>) (Fig. S2), which are traditional markers for anthropogenic sources. The  $C_{18:0}/C_{16:0}$  ratio can be used to determine the source of fatty acids. A value of the  $C_{18:0}/C_{16:0}$  ratio of <0.25 denotes herb/wood burning and terrestrial plant waxes, a value of 0.25–0.50 indicates fossil fuel combustion, and a value of 0.50–1.00 suggests kitchen emissions and road dust (Ren et al., 2016). In this study, the value of the  $C_{18:0}/C_{16:0}$  ratio was in the range 0.44–1.58 (0.62  $\pm$  0.21), suggesting that the fatty acids in winter in Chengdu derived mainly from anthropogenic activities. As validation of the

discussion, the correlation analysis found that fatty acids showed strong correlation with the tracers of biomass burning or fossil fuel combustion, e.g., levoglucosan ( $R^2 = 0.41$ ), hopanes ( $R^2 = 0.64$ ), and PAHs ( $R^2 = 0.50$ ).

In urban environments, unsaturated fatty acids can be derived from anthropogenic sources such as cooking, vehicular emissions, and biomass burning (Rogge et al., 1996), and they can be rapidly oxidized once emitted to the atmosphere. In this study, the average concentration of  $C_{18:1}$  (oleic acid) was  $143.1 \pm 274.6$  ng m<sup>-3</sup>, and its concentration at nighttime ( $251.9 \pm 357.6$  ng m<sup>-3</sup>) was 7.34 times higher than that during daytime ( $34.3 \pm 28.7$  ng m<sup>-3</sup>), which can be attributed to its rapid oxidation during daytime. Conversely,  $C_{18:2}$  (linoleic acid) was not detected in our study, indicating that  $C_{18:2}$  degrades more quickly than  $C_{18:1}$  owing to the two double bonds in  $C_{18:2}$ . According to the evaluation criteria, namely: wood combustion, leaf surfaces of plants, and meat cooking operations with  $C_{18:2} > C_{18:1}$ , the detected concentration of  $C_{18:1}$  reflected vehicular emissions (Rogge et al., 1993). We can infer that vehicular emissions have substantial impact on the fatty acids found in Chengdu. The  $C_{18:0}/C_{18:1}$  ratio was  $2.95 \pm 1.93$  in daytime versus  $2.38 \pm 2.31$  at nighttime, suggesting enhanced photochemical degradation of unsaturated fatty acids during daytime (Fu et al., 2008).

Fatty alcohols. Owing to their low content, fatty alcohols with odd carbon numbers were detected only in a very small number of samples. Therefore, this study focused on those fatty alcohols with even carbon numbers ( $C_{20}$ – $C_{32}$ ) that were detected with high contents in most of the samples. The average concentration of fatty alcohols was 54.1 ± 25.2 ng m<sup>-3</sup>, and the nighttime concentration ( $60.0 \pm 25.9 \text{ ng m}^{-3}$ ) was higher than the daytime concentration ( $48.2 \pm 23.4 \text{ ng m}^{-3}$ ). The concentration determined in this study is lower than that observed in winter in Chennai ( $67.9 \pm 26.6 \text{ ng m}^{-3}$ ) (Fu et al., 2010) and in Tianjin ( $1310 \pm 811 \text{ ng m}^{-3}$  during daytime,  $1520 \pm 1010 \text{ ng m}^{-3}$  at nighttime) (Fan et al., 2020), but higher than that observed in a marine region (0.07– $8.3 \text{ ng m}^{-3}$ ) (Fu et al., 2011). Previous studies highlighted that HMW (> $C_{19}$ ) fatty alcohols are abundant in higher plant waxes and loess deposits (Simoneit et al., 1991), and they might also be emitted to the air via biomass burning (Fu et al., 2008). The substantial contribution of biomass burning in our study area can be demonstrated by the strong correlation between fatty alcohols and levoglucosan ( $R^2 = 0.64$ ).

#### 3.2.2 Sugar compounds







Anhydrosugars. Levoglucosan, mannosan, and galactosan have been used as key tracers for biomass burning emissions (Simoneit et al., 1999). They are emitted exclusively via the combustion and pyrolysis of cellulose and hemicelluloses. In this study, the concentration range of anhydrosugars was 44.3-791.3 ng m<sup>-3</sup>, with the average of  $362.3 \pm 162.1$  ng m<sup>-3</sup>, which is substantially higher than the background level in winter in southwestern China, e.g., Mt. Gongga ( $120 \pm 61.3$  ng m<sup>-3</sup>) (Cui et al., 2023). The concentration of levoglucosan was  $239.8 \pm 100.1$  ng m<sup>-3</sup>, which is substantially higher than that found in mountain areas ( $111 \pm 46.3$  ng m<sup>-3</sup>) (Cui et al., 2023), forests (112 ng m<sup>-3</sup>) (Fu et al., 2010), marine areas (2.9 ng m<sup>-3</sup>) (Fu et

al., 2011), the Arctic Ocean (0.37 ng m<sup>-3</sup>) (Fu et al., 2013), and the inner Tibetan Plateau (19.2  $\pm$  9.19 ng m<sup>-3</sup>) (Wan et al., 2023). Meanwhile, it is comparable with the value found in winter in Xi'an (268.5 ng m<sup>-3</sup>) (Wang et al., 2018c), but much lower than the level observed in Chengdu in 2011 (635  $\pm$  246 ng m<sup>-3</sup>) (Tao et al., 2014). This indicates that policies implemented in recent years to control biomass burning have had positive effects in the SCB. Additionally, the higher concentration at nighttime (258.9  $\pm$  102.4 ng m<sup>-3</sup>) than during daytime (220.7  $\pm$  95.5 ng m<sup>-3</sup>) is consistent with results reported for Tianjin (296  $\pm$  153 ng m<sup>-3</sup> at nighttime, 205  $\pm$  122 ng m<sup>-3</sup> during daytime) (Fan et al., 2020).

Although the concentration of both mannosan ( $52.5 \pm 29.0 \text{ ng m}^{-3}$ ) and galactosan ( $70.0 \pm 36.2 \text{ ng m}^{-3}$ ) was much lower than that of levoglucosan, both exhibited very strong correlation with levoglucosan ( $R^2 = 0.83$  and 0.86, respectively). Meanwhile, the concentration of galactosan was higher than that of mannosan during both daytime and nighttime, consistent with the results of research on smoke particles derived from rice straw, biomass briquettes, and grasses (Oros et al., 2006). Although levoglucosan dominated the total sugars in our study, its contribution (57.5%) is lower than that reported 2003 in winter in Beijing (86.4%), Guangzhou (87.5%), and another megacity in the SCB, i.e., Chongqing (96.7%) (Wang et al., 2006). This suggests that after reducing biomass burning emissions, the chemical composition of sugars has changed.

Levoglucosan is largely produced by thermal decomposition of cellulose, while mannosan is mainly a product of pyrolysis of hemicellulose (Simoneit et al., 1999). Therefore, different types of biomass burning can be distinguished using the levoglucosan/mannosan (L/M) ratio because the ratio varies markedly in relation to the burning of softwood (2.5–6.7), hardwood (12.9–35.4), and agricultural residues (22.6–55.7) (Wan et al., 2019). In this study, the value of the L/M ratio was in the range 2.83–17.63 (5.40 ± 2.34), which indicates that the major contribution was from softwood burning. The L/M ratio in Chengdu aerosols is comparable with that found in central Alaska (4.6) (Deshmukh et al., 2019) and in Tianjin (7.38) (Fan et al., 2020), but much lower than that reported for inner parts of the Tibetan Plateau (13.8) (Wan et al., 2023), Dushanbe (12.1) (Chen et al., 2022), and rural areas in northwest China (13.7) (Liu et al., 2024), where hardwoods and crop residues are the major sources of biomass burning.

Water-soluble potassium ( $K^+$ ) is used widely as a tracer of biomass burning. Accordingly, we found that  $K^+$  showed strong correlation with levoglucosan ( $R^2$  = 0.71) in our study, which is different from the results obtained in other regions, where the correlation between  $K^+$  and levoglucosan was weak (Chen et al., 2022; Cui et al., 2023). This can be attributed to other sources making strong contributions to the concentrations of these two substances, such as dust, sea salt, cooking, and coal combustion for  $K^+$ , and municipal solid waste burning, burning of fireworks, and meat cooking for levoglucosan (Wu et al., 2021). When the contributions of these sources were excluded,  $K^+$  and levoglucosan showed stronger correlation (Chen et al., 2022; Cui et al., 2023). Meanwhile, the photodegradation of levoglucosan by free radicals in the atmosphere can also weaken the correlation

between them (Deshmukh et al., 2019). The strong correlation between these two tracers in our study means limited contribution by other sources. For example, the weak contribution of dust for  $K^+$  can be proven by the weak correlation between  $K^+$  and  $Ca^{2+}$  ( $R^2 = 0.15$ ). Meanwhile, the temperature in winter was low (7.2  $\pm$  3.9 °C), resulting in weak photolysis of levoglucosan. Therefore, compared with other regions,  $K^+$  and levoglucosan had more similar sources in winter in Chengdu, i.e., biomass burning.






**Primary saccharides**. There were four primary saccharides (i.e., fructose, glucose, sucrose, and trehalose) identified in our study, and their total concentration ( $43.1 \pm 16.8 \text{ ng m}^{-3}$ ) is close to that reported in winter in Tianjin ( $46.8 \pm 20.9 \text{ ng m}^{-3}$  during daytime,  $49.9 \pm 23.0 \text{ ng m}^{-3}$  at nighttime) (Fan et al., 2020), but much lower than that found in the area of Mt. Gongga ( $138 \pm 31.5 \text{ ng m}^{-3}$ ). Fructose was the predominant species ( $28.5 \pm 13.8 \text{ ng m}^{-3}$ ), followed by glucose ( $11.4 \pm 3.6 \text{ ng m}^{-3}$ ), sucrose ( $2.4 \pm 1.0 \text{ ng m}^{-3} \text{ ng m}^{-3}$ ), and trehalose ( $0.9 \pm 0.9 \text{ ng m}^{-3}$ ). The sources of primary saccharides are diverse. For example, trehalose is emitted from soil dust, microbial, and fungal spores (Wang et al., 2021); sucrose is emitted from pollen, spores, and dust (Wan et al., 2019); and glucose and fructose are derived predominantly from plant debris and sometimes from microorganisms, soil dust, and biomass burning (Simoneit et al., 2004).

Generally, strong winds, high ambient temperature, and convective activity during daytime promote the release of pollen and lead to enhanced entrainment and dispersal of particles into the atmosphere. However, in our study, the concentration of primary saccharides showed higher concentrations at nighttime ( $45.4 \pm 17.8 \text{ ng m}^{-3}$ ) than during daytime ( $40.8 \pm 15.5 \text{ ng m}^{-3}$ ). Therefore, the promotional effect of favorable weather conditions during daytime reported in previous studies might not have been relevant in Chengdu. Accordingly, primary saccharides displayed no correlation with ambient temperature in our study ( $R^2 = 0.007$ ). This might be because the temperature during the study period remained at a relatively low level ( $7.2 \pm 3.9 \,^{\circ}$ C), meaning that it had only weak impact on the concentration of primary saccharides; other factors such as changes in the boundary layer height might have had greater impact.

**Sugar alcohols.** There were three sugar alcohol compounds were detected: arabitol, mannitol, and inositol. The mean concentration of the sugar alcohols was  $11.7 \pm 5.4$  ng m<sup>-3</sup>, and the concentration at nighttime ( $12.6 \pm 5.6$  ng m<sup>-3</sup>) was a bit higher than that during daytime ( $10.8 \pm 5.0$  ng m<sup>-3</sup>). Zhu et al. (2016) attributed this daily variation to higher relative humidity at night, which causes enrichment of fungal spores and their discharges, resulting in higher abundances of their tracers in the aerosols. Correspondingly, in this study, the relative humidity at nighttime ( $85.4 \pm 12.6\%$ ) was higher than that observed during daytime ( $71.1 \pm 13.1\%$ ). Additionally, Graham et al. (2003) hypothesized that enhanced nighttime concentrations of sugar alcohols might be associated with the observed nocturnal increase in yeasts and other small fungal spores.

Arabitol was the most abundant species  $(5.3 \pm 2.9 \text{ ng m}^{-3})$  and it accounted for 45.5% of the total concentration of sugar

alcohols. We found strong correlation among the three sugar alcohol compounds ( $R^2 = 0.49-0.57$ ), which implies that they had similar sources. This is different from the results reported by Wang et al. (2021), for forest areas in southwestern China. They found poor correlation between mannitol and arabitol (r = 0.38) but strong positive correlation between trehalose and mannitol (r = 0.79), indicating that notable differences might exist in the sources of sugar alcohol compounds between city and forest areas. In addition, the strong correlation was found between mannitol and inositol with levoglucosan ( $R^2 = 0.57$  and 0.61, respectively), indicating the significant contribution of biomass burning to sugar alcohol compounds.

#### 3.2.3 Hopanes







Hopanes are considered to be derived from fossil fuel combustion, especially that associated with vehicular emissions and coal combustion (Wang et al., 2009). In this study, a number of hopanes ( $C_{27}$ – $C_{32}$  except for  $C_{28}$ ) were detected with an average concentration of 1.3 ± 0.7 ng m<sup>-3</sup>, and the concentration during daytime (1.2 ± 0.6 ng m<sup>-3</sup>) was close to that observed at nighttime (1.4 ± 0.9 ng m<sup>-3</sup>). The observed concentration is much lower than that found in winter in certain other cities, e.g., 14.4 ± 9.09 ng m<sup>-3</sup> in Chennai (Fu et al., 2010), 56.3 ng m<sup>-3</sup> in Beijing (Wang et al., 2020), and 7.21 ± 6.20 ng m<sup>-3</sup> in Nanjing (Cao et al., 2021), but close to the results found in summer in some mountain areas, e.g., 1.2–1.6 ng m<sup>-3</sup> in the Mt. Tai region (Fu et al., 2012). The  $29\alpha\beta/30\alpha\beta$  ratio of coal combustion sources is generally >1, whereas that associated with traffic emissions is <1 (Tian et al., 2021). In this study, the mean  $29\alpha\beta/30\alpha\beta$  ratio was 0.87. Meanwhile, the value of the  $31\alpha\beta S/(31\alpha\beta S + 31\alpha\beta R)$  ratio (0.57) is closer to the ratio of mineral oil-derived sources (e.g., vehicular exhausts) (0.57–0.59) (Schnellekreis et al., 2005). All this evidence suggests that vehicular emissions make a substantial contribution to the hopanes found in the aerosols in this megacity during winter.

#### 3.2.4 PAHs and OPAHs

The presence of PAHs, which are persistent organic pollutants that result from incomplete combustion of hydrocarbon-containing substances (such as biomass, fossil fuels, and some natural factors, e.g., wildfires) (Zheng et al., 2020), represents a health concern owing to their carcinogenicity, genotoxicity, and potential endocrine disruptiveness (Fu et al., 2010). In this study, 15 PAHs (3-ring to 7-ring) were detected, with an average concentration of  $13.3 \pm 8.2$  ng m<sup>-3</sup>. This is comparable with the level reported for winter in Shanghai ( $15.54 \pm 5.56$  ng m<sup>-3</sup>) (Cao et al., 2021) and Ngari on the Tibetan Plateau (11.5 ng m<sup>-3</sup>) (Zheng et al., 2020), lower than the level reported for Chennai ( $35.7 \pm 18.7$  ng m<sup>-3</sup>) (Fu et al., 2010), Greater Cairo (58.9 ng m<sup>-3</sup>) (Farah et al., 2024), Nanjing ( $37.8 \pm 12.0$  ng m<sup>-3</sup>) (Cao et al., 2021), Zhengzhou (120 ng m<sup>-3</sup>) (Dong et al., 2024), and Beijing (317.1 ng m<sup>-3</sup>) (Wang et al., 2020), but much higher than the level reported for marine regions (0.20 ng m<sup>-3</sup>) (Fu et al., 2011). Additionally, the concentration of PAHs at nighttime ( $15.8 \pm 9.7$  ng m<sup>-3</sup>) was 1.48 times higher than that found during daytime ( $10.7 \pm 5.5$  ng m<sup>-3</sup>). This might reflect the lower height of the boundary layer and poorer mixing at night, but it also

might reflect the fact that PAHs tend to condense to the particulate phase at night because of the lower temperatures.







The ratio of different PAHs is considered an important indicator for determining their sources. For example, the IP/BghiP concentration ratio is 0.2, 0.5, and 1.3 in the smoke from gasoline, diesel, and coal combustion, respectively; the BghiP/BeP concentration ratio is 2.0 and 0.8 in vehicular exhaust and coal combustion emissions, respectively; and the BaP/BeP concentration ratio is lower (higher) than 0.6 for nontraffic (traffic) sources (Wang et al., 2009; Li et al., 2019a). In this study, the values of the IP/BghiP ( $1.10 \pm 0.13$ ), BghiP/BeP ( $1.13 \pm 0.16$ ), and BaP/BeP (0.73) ratios were between those representing vehicular exhaust and coal combustion emissions, suggesting that the PAHs in Chengdu in winter largely originated from the mixed emissions of these two sources. Accordingly, we found strong correlation between PAHs and hopanes ( $1.00 \pm 0.00$ ) Meanwhile, the relative contributions of LMW ( $1.00 \pm 0.00$ ), and HMW ( $1.00 \pm 0.00$ ), and  $1.00 \pm 0.00$ 0 of the total concentration of PAHs during clear and haze periods, respectively (Li et al., 2021b). One important reason for the difference might be the different atmospheric temperatures in winter between the two cities. The temperature in Jinan ( $1.00 \pm 0.00$ ) in the clear and haze periods, respectively is much lower than that in Chengdu ( $1.00 \pm 0.00$ ), and lower temperatures promote the partitioning of LMW PAHs to the particulate phase.

OPAHs are not only emitted into the atmosphere simultaneously with PAHs through similar combustion processes, but are also generated through homogeneous or heterogeneous reactions between parent PAHs and atmospheric oxidants (e.g., O<sub>3</sub> and OH) (Shin et al., 2022). It has been suggested that OPAHs represent one group of the species fundamental in the formation of reactive oxygen species (Chung et al., 2006; Cassee et al., 2013). Therefore, they could cause oxidative damage to biological molecules, such as DNA and proteins, thereby representing a more toxic hazard (10-100,000 times) than PAHs (Lin et al., 2015; Wang et al., 2006). There were 11 OPAHs detected in the Chengdu aerosols, and their total average concentration was  $43.0 \pm 30.0$  ng m<sup>-3</sup>. Similar to the PAHs, the OPAHs were found to have higher concentrations at nighttime (53.4  $\pm$  35.0 ng m<sup>-3</sup>) than during daytime (32.6 ± 19.5 ng m<sup>-3</sup>). The observed concentration level is much higher than that found in winter in Nanjing (12.9 ng m<sup>-3</sup> during daytime, 15.5 ng m<sup>-3</sup> at nighttime) (Miettinen et al., 2019) and in Seoul (9.40  $\pm$  2.50 ng m<sup>-3</sup>) (Shin et al., 2022), but lower than the result reported for winter in Beijing (61 ng m<sup>-3</sup>) (Ren et al., 2024), especially during heavy pollution periods (581.4 ± 299.8 ng m<sup>-3</sup>) (Li et al., 2019a). Meanwhile, the concentration of OPAHs was 3.24 times greater than that of the PAHs. This ratio is much higher than the results reported for summer (1.06) and winter (0.81) in Nanjing (Miettinen et al., 2019) and in summer (0.7) and winter (0.5) in Xi'an (Bandowe et al., 2014), highlighting the stronger emission or secondary generation of OPAHs in our study. Although biomass burning is considered an important source of OPAHs (Miettinen et al., 2019), the weak correlation of the OPAHs with levoglucosan ( $R^2 = 0.26$ ) and  $K^+$  ( $R^2 = 0.08$ ) suggests that the contribution from biomass burning to the OPAHs in Chengdu was limited, and that secondary processes might be the more

important source. Lin et al. (2015) also found that secondary formation contributed substantially (53.3% during the non-heating period) to the concentration of OPAHs in winter in Beijing. Another study in Beijing also reported that secondary formation accounted for 8.9%–99% (average: 73%) of the concentration of OPAHs during autumn and winter, but this is almost the opposite to that observed in Seoul, where OPAHs were heavily influenced by primary emissions (Shin et al., 2022).

#### 3.2.5 Phthalic acids







Generally, phthalic acids are considered secondary oxidation products of PAHs (Fine et al., 2004) that can play an important role in enhancing new atmospheric particle formation (Zhang et al., 2004). In our study, three phthalic acids (i.e., o-, m-, and p-isomers) were detected; their total average concentration was  $68.9 \pm 34.2$  ng m<sup>-3</sup>, and the concentration at nighttime ( $74.6 \pm 34.6$  ng m<sup>-3</sup>) was higher than that during daytime ( $63.1 \pm 33.3$  ng m<sup>-3</sup>). Correlation analysis revealed positive correlation between the concentrations of phthalic acids and PAHs at nighttime ( $R^2 = 0.36$ ), but almost no correlation during daytime ( $R^2 = 0.07$ ). Meanwhile, although OPAHs are also secondary reaction products of PAHs, the correlation between the OPAHs and the phthalic acids was weak ( $R^2 = 0.12$ ), suggesting that they underwent different secondary reaction processes. Further analysis revealed that the correlation between the phthalic acids and  $SO_4^{2-}$  ( $R^2 = 0.36$ ) was stronger than that between the OPAHs and  $SO_4^{2-}$  ( $R^2 = 0.17$ ); however, no correlation was found between the phthalic acids and  $NO_3^{--}$  ( $R^2 = 0.02$ ), and the OPAHs showed negative correlation with  $NO_3^{--}$  ( $R^2 = 0.24$ ). Additionally, differences in the primary sources might also explain the weak correlation between the OPAHs and the phthalic acids. For example, we found that the correlation between the phthalic acids and levoglucosan ( $R^2 = 0.61$ ) was much stronger than that between the OPAHs and levoglucosan ( $R^2 = 0.26$ ). These analysis once again proves the complexity of the primary emission and secondary generation of atmospheric OA.

#### 3.2.6 Phthalate esters

Phthalate esters, used widely as plasticizers in synthetic polymers or as softeners in polyvinylchlorides, can be emitted into the atmosphere via evaporation (Fu et al., 2012). They have potential adverse effects on the ecological system and human health owing to their toxicity (Fu et al., 2010). In our study, three phthalate esters were detected: diisobutyl (DiBP), di-n-butyl (DnBP), and bis(2-ethylhexyl) phthalates. The total average concentration of phthalate esters was  $571.8 \pm 246.7 \text{ ng m}^{-3}$ , which is much higher than that reported for rural areas ( $140 \pm 62.1 \text{ ng m}^{-3}$ ) (Liu et al., 2024), forests ( $303 \text{ ng m}^{-3}$ ) (Fu et al., 2010), and the Arctic Ocean ( $2.6 \text{ ng m}^{-3}$ ) (Fu et al., 2013). The higher concentration during daytime ( $622.7 \pm 253.2 \text{ ng m}^{-3}$ ) than that found at nighttime ( $520.8 \pm 232.6 \text{ ng m}^{-3}$ ) might reflect enhanced emission of phthalate esters from plastics owing to the higher ambient temperature and intensity of production and life during daytime. For example, Ge et al. (2024) found that plasticizer-related OA were associated with increased traffic activity, tire wear, and coal combustion. The correlation between the phthalate esters and levoglucosan was weak (82 = 0.27), which means that biomass burning was not an important contributor

to this type of OA compound.

## 3.2.7 Nitrophenols

Nitrophenols can be either emitted directly via biomass burning or formed in the atmosphere through gas-phase and aqueous-phase reactions of aromatic precursors (Lu et al., 2011; Xie et al., 2019). In this study, nine nitrophenols were detected. The average value of their total concentration was  $5.2 \pm 3.8$  ng m<sup>-3</sup>, which is much lower than the concentration reported for winter in Jinan ( $48 \pm 26$  ng m<sup>-3</sup>) (Wang et al., 2018b) and that found before the heating period ( $20 \pm 21$  ng m<sup>-3</sup>) and during the heating period ( $53 \pm 51$  ng m<sup>-3</sup>) in Beijing (Ren et al., 2024), but comparable with that of the background mountain atmosphere in the Mt. Wuyi ( $3.9 \pm 1.5$  ng m<sup>-3</sup>) (Ren et al., 2023). This variation in nitrophenols is linked to emission sources, formation pathways, and weather conditions (Ren et al., 2024). Despite the differences in concentration levels, similar to previous research (Ren et al., 2024), 4-nitrocatechol (4NC) and 4-nitrophenol (4NP) were found to be the dominant nitrophenol species, contributing 42.4% and 39.2% to the total nitrophenols mass, respectively.

# 3.2.8 Biogenic SOA tracers



Globally, emissions of biogenic VOCs (1150 Tg C yr<sup>-1</sup>) are much higher than emissions of anthropogenic VOCs (110 Tg C yr<sup>-1</sup>) (Guenther et al., 2006). In this study, the concentration of total biogenic SOA tracers was in the range 12.1–329.2 ng m<sup>-3</sup> (116.9 ± 61.2 ng m<sup>-3</sup>). Not only is this higher than the level observed in some urban areas, e.g., 27.03 ng m<sup>-3</sup> during daytime and 26.24 ng m<sup>-3</sup> at nighttime in Tianjin (Fan et al., 2020), 58.6 ng m<sup>-3</sup> in Guangzhou, and 39.92 ng m<sup>-3</sup> in Zhuhai (Zhang et al., 2019), but it is also higher than that observed in some remote background areas, e.g., Namco (2.30 ± 1.09 ng m<sup>-3</sup>) (Wan et al., 2023), Alaska (4.64 ng m<sup>-3</sup>) (Haque et al., 2016), and Mt. Gongga (39.4 ± 18.5 ng m<sup>-3</sup>) (Cui et al., 2023). This emphasizes the important contribution of biogenic SOA to the OA in winter in Chengdu.

**Isoprene SOA tracers (SOA<sub>I</sub>).** Isoprene has conjugated double bonds, meaning that it is more reactive toward oxidants such as  $O_3$  and NOx, and more prone to creating various intermediates and stable products (Fu et al., 2010). In this study, we detected eight  $SOA_I$ , i.e., 2-methylglyceric acid (2-MGA), three C5-alkene triols (*cis*-2-methyl-1,3,4-trihydroxy-1-butene, 3-methyl-2,3,4-trihydroxy-1-butene, and *trans*-2-methyl-1,3,4-trihydroxy-1-butene), two 2-methyltetrols (2-methylthreitol and 2-methylerythritol, 2-MTLs), and *cis*- and *trans*-3-methyltetrahydrofuran-3,4-diol (3-MeTHF-3,4-diols). The total average concentration of  $SOA_I$  was  $6.7 \pm 3.3$  ng m<sup>-3</sup>.

As the major fractions of SOA<sub>I</sub>, the C5-alkene triols, 2-MTLs, and 3-MeTHF-3,4-diols are higher-generation products from the photooxidation of epoxydiols of isoprene under low-NOx conditions (Cui et al., 2023; Surratt et al., 2010). However, 2-MGA is formed mainly under high-NOx conditions (Cui et al., 2023). Therefore, NOx might have substantial impact on the

chemical composition of  $SOA_1$ . In Chengdu, the ambient  $NO_3^-$  and  $NO_2$  concentrations in winter were  $25.1 \pm 12.8$  and  $40.9 \pm 15.3$  µg m<sup>-3</sup>, respectively, which might promote massive generation of 2-MGA, while limiting the generation of C5-alkene triols, 2-MTLs, and 3-MeTHF-3,4-diols. This can be further confirmed by the 2-MGA/2-MTLs ratio, which is frequently used to reveal the impact of NOx on  $SOA_1$  formation. The 2-MGA/2-MTLs ratio in this study was  $3.42 \pm 1.44$ , which is higher than that reported for Alaska (0.12) (Deshmukh et al., 2019), Mt. Gongga (0.07) (Cui et al., 2023), Guangzhou (0.46), Zhuhai (0.73), and Dongguan (0.35) (Zhang et al., 2019). Additionally, the higher C5-alkene triols/2-MTLs concentration ratio (9.06) is substantially different to the low ratio obtained in laboratory experiments for isoprene photooxidation in the absence of NOx (<0.10) (Kleindienst et al., 2009). All this evidence emphasizes the notable impact of high concentrations of NOx on the chemical composition of  $SOA_1$  in Chengdu. Moreover, although 2-MGA and 2-MTLs are formed under different NOx conditions, they both exhibited positive correlation with relative humidity ( $R^2 = 0.39$  and 0.36, respectively). This is different from the results reported by Shen et al. (2015), they found that low relative humidity (15%-40%) could enhance the formation of 2-MGA but not that of 2-MTLs. It can be seen that relative humidity plays an important role in promoting the formation of SOA<sub>1</sub> in Chengdu under two different NOx environments. Furthermore, aerosol acidity is believed to effectively enhance the generation of  $SOA_1$ . Accordingly, we found strong positive correlation of  $SOA_1$  with  $SOA_2^2-(R^2 = 0.66)$  and  $NOA_3-(R^2 = 0.71)$ .







Monoterpene SOA tracers (SOA<sub>M</sub>). Four organic acids were detected as  $\alpha/\beta$ -pinene oxidation products: pinic acid (PA), pinonic acid (PNA), 3-hydroxyglutaric acid (3-HGA), and 3-methyl-1,2,3-butanetricarboxylic acid (MBTCA). The total average concentration of SOA<sub>M</sub> was  $39.6 \pm 20.3$  ng m<sup>-3</sup>. As a first-generation oxidation product, the PA concentration (3.5  $\pm$ 2.1 ng m<sup>-3</sup>) was 5.8 times greater than that of PNA ( $0.6 \pm 0.3$  ng m<sup>-3</sup>) because the vapor pressure of PA is approximately two orders of magnitude lower than that of PNA. Biomass burning was found to have an important impact on PA and PNA, as supported by the good correlation of levoglucosan with PA ( $R^2 = 0.68$ ) and PNA ( $R^2 = 0.43$ ). As a high-generation photooxidation product of  $\alpha$ -pinene, 3-MBTCA was formed via oxidation of the first-generation SOA<sub>M</sub>, such as PNA and PA. Thus, the (PA + PNA)/MBTCA ratio can be used to understand SOA<sub>M</sub> aging. In fresh chamber-produced α-pinene SOA samples, (PA + PNA)/MBTCA ratio values were found in the range 1.51–3.21 (Offenberg et al., 2007). In our study, the (PA + PNA)/MBTCA ratio was 0.23 ± 0.16, which is much lower than that reported for Namco (4.83 during daytime, 4.59 at nighttime) (Wan et al., 2023), Alaska (21.0) (Haque et al., 2016), and Tianjin (17.6 during daytime, 31.0 at nighttime) (Fan et al., 2020). This suggests that the SOA<sub>M</sub> in Chengdu were at a very high level of aging. Additionally, good correlations were obtained between the biomass burning tracer, i.e., levoglucosan, and the higher-generation oxidation products (e.g., 3-HGA,  $R^2 = 0.83$ ). Therefore, it is suggested that SOA<sub>M</sub> might be affected substantially by biomass burning activities. Jaoui et al. (2005) reported that α-pinene SOA has considerably higher yields of 3-MBTCA relative to 3-HGA than those of β-pinene. Therefore, the 3-HGA/3-MBTCA ratio can be used to distinguish the contribution of precursor compounds for SOA<sub>M</sub>. For example, Lewandowski et al. (2013) found lower values of the 3-HGA/3-MBTCA ratio in the southeastern U.S. ( $\sim$ 1.0), and they suggested that  $\alpha$ -pinene was the major precursor for SOA<sub>M</sub>. The ratio in the Mt. Gongga region was only 0.31, suggesting a large contribution of  $\alpha$ -pinene to SOA<sub>M</sub> formation (Cui et al., 2023). In this study, the 3-HGA/3-MBTCA ratio was 0.71, suggesting greater contribution by  $\alpha$ -pinene oxidation products than by  $\beta$ -pinene oxidation products in the formation of SOA<sub>M</sub>.

Sesquiterpene SOA tracers (SOAs). β-Caryophyllinic acid (β-CPA) is a photooxidation or ozonolysis product of β-caryophyllene (Cui et al., 2023; Fu et al., 2011). In our study, the concentration of β-CPA was  $70.6 \pm 42.2$  ng m<sup>-3</sup>, which is much higher than the observation results of background stations in the same region, i.e.,  $27.3 \pm 16.8$  ng m<sup>-3</sup> in Mt. Gongga (Cui et al., 2023). Akagi et al. (2011) reported that biogenic VOCs could be emitted via biomass burning. Haque et al. (2016) also attributed elevated β-CPA concentrations during haze periods to biomass burning. Accordingly, we found that all three types of biogenic SOA tracers showed strong positive correlation with levoglucosan (R<sup>2</sup> = 0.55–0.69). This is different from observations in cities in northern China. For example, Haque et al. (2016) found statistically significant positive correlation between β-CPA and levoglucosan (r = 0.65), whereas no correlation was observed between levoglucosan and SOA<sub>I</sub> (r = -0.18) or a/β pinene SOA tracers (r = -0.42). These results not only highlight the important contribution of biomass burning to biogenic SOA tracers in Chengdu, but also reveal the spatiotemporal differences in the sources of them.

Comparison of the daytime and nighttime results obtained in this study revealed that biogenic SOA have similar or higher concentrations at nighttime relative to those found during daytime, and that there is no correlation between the three types of biogenic SOA and T ( $R^2 < 0.04$ ). This finding is different to that of many previous studies that reported higher levels of biogenic SOA during daytime than at nighttime, and that found their concentrations showed strong correlation with temperature (especially for SOA<sub>I</sub> and SOA<sub>M</sub>) (Fu et al., 2012; Wan et al., 2023; Cui et al., 2023). Therefore, the sources and formation mechanisms of biogenic SOA in winter in Chengdu might differ substantially from those found in other regions.

# 3.3 OC source analysis







#### 3.3.1 Tracer-based methods

The contribution of each source to the total OC was evaluated using a tracer-based method. The contribution of biomass burning was calculated using the ratio of levoglucosan to OC (8.14%) detected in the source samples (Wan et al., 2017). The contribution of fungal spores was estimated based on experimental values of 1.7 pg of mannitol and 13 pg of OC per spore (Bauer et al., 2008). The contribution of plant debris was evaluated using the tracer of glucose and an experimentally derived factor (Puxbaum, 2003). The tracer mass fractions of isoprene (0.155  $\pm$  0.039),  $\alpha$ -pinene (0.231  $\pm$  0.111),  $\beta$ -caryophyllene (0.023  $\pm$  0.005), and phthalic acid (0.0199  $\pm$  0.0084) were applied to estimate the contributions of isoprene,  $\alpha$ -pinene,  $\beta$ -caryophyllene, and anthropogenic secondary OC (SOC), respectively (Kleindienst et al., 2007). Although there might be some

uncertainty inherent in the practical application of the above estimation methods owing to differences between the real atmosphere and smog chamber experiments, a broad estimation of the contribution from each source to the total OC can be deduced (Wu et al., 2020).

We found that biomass burning OC, as one of the important sources of OC, contributed 10.0% to the total OC (Table 1). The other two primary sources, namely plant debris OC and fungal spores OC, had extremely low contributions of only 0.10% and 0.08%, respectively. In terms of SOC sources, the contributions of isoprene SOC and  $\alpha$ -pinene SOC were 0.15% and 0.57%, respectively, i.e., much lower than the contribution of  $\beta$ -caryophyllene SOC (11.0%). Additionally, with a contribution of 11.6%, anthropogenic SOC was found to be the most important source of OC.

Table 1: Contributions of OC from primary sources and secondary formation to OC in different studies.





| OC sources          | Chengdu      | Bode <sup>a</sup> | Nam Co <sup>b</sup> | Alaska <sup>c</sup> | Tianjin (day vs | Arctic | Wakayama            | Changchung |
|---------------------|--------------|-------------------|---------------------|---------------------|-----------------|--------|---------------------|------------|
|                     | (This study) |                   |                     |                     | night)d         | Oceane | Forest <sup>f</sup> |            |
| Biomass burning OC  | 10.0%        | 27.9%             | 15.4%               |                     | 12.1%, 16.0%    |        | 0.5%, 0.2%          | 27%        |
| Plant debris OC     | 0.10%        | 0.17%             | 0.5%                |                     | 0.10%, 0.08%    |        | 4.6%, 5.6%          |            |
| Fungal spores OC    | 0.08%        | 0.24%             | 1.7%                |                     | 0.19%, 0.18%    | 10.7%  | 22%, 45%            | 0.2%       |
| Isoprene SOC        | 0.15%        | 0.61%             | 0.4%                | 0.08%               | 0.19%, 0.16%    | 3.8%   | 13%, 10%            | 0.05%      |
| α-pinene SOC        | 0.57%        | 0.39%             | 0.3%                | 0.63%               | 0.31%, 0.30%    | 2.9%   | 5.1%, 3.7%          | 0.05%      |
| β-caryophyllene SOC | 11.0%        | 0.52%             | 0.5%                | 9.6%                | 2.31%, 2.25%    | 0.19%  | 0.8%, 0.7%          | 0.3%       |
| Anthropogenic SOC   | 11.6%        | 5.34%             |                     |                     |                 |        |                     | 3.9%       |

a (Wan et al., 2019), b (Wan et al., 2023), c (Haque et al., 2016), d (Fan et al., 2020), e (Fu et al., 2013), f (Zhu et al., 2016), and g (Wu et al., 2020).

Table 1 lists the results of our study with those of previous studies on other cities or background areas. It can be seen that the contribution of biomass burning OC in our study is much lower than that reported in previous studies (except for the forest site). The contributions of plant debris, fungal spores, isoprene, and  $\alpha$ -pinene OC or SOC are much lower than those reported for the Arctic Ocean (Fu et al., 2013) and forest site (Zhu et al., 2016), but comparable with the values found in other regions. However, the contribution of  $\beta$ -caryophyllene SOC is the highest among all the studies, even reaching the level of one order of magnitude higher than the results reported for Changchun (Wu et al., 2020) and some background areas. Moreover, the contribution of anthropogenic SOC is also higher than that found in Bode (Wan et al., 2019) and Changchun (Wu et al., 2020).

The relative contributions of daytime and nighttime OC sources were very close (Fig. S3). In comparison with daytime values, we found that only the contributions of biomass burning, β-caryophyllene, and anthropogenic OC or SOC increased by 0.8%, 1.1%, and 0.8%, respectively in nighttime, while the diurnal differences in the contributions of other OC sources could be considered negligible. The slight daytime–nighttime difference was similar to that observed in winter in Tianjin (Fan et al., 2020), but markedly different from the observation results derived in mid-latitude forest regions, where the largest daytime–nighttime difference occurred in fungal spores OC (23% higher at nighttime than during daytime) (Zhu et al., 2016). The

composition and diurnal differences of OC sources in different studies are closely related to various factors, such as emission sources, meteorological conditions, volatility of organic compounds, and regional transmission. For example, based on their finding that the large biogenic SOC fraction occurred mainly on days with anthropogenic influences, whereas on days closer to natural conditions, biogenic SOC made a smaller contribution, Zhu et al. (2016) suggested that increased fossil fuel combustion and the subsequent oxidant would cause a larger contribution of biogenic SOC to the total OC.

## 3.3.2 PMF source apportionment







Seven OC sources were identified using PMF, i.e., coal combustion, vehicular emissions, biomass burning, primary biogenic emissions, dust, plastic related sources, and secondary formation (S3). As shown in Fig. 2, secondary formation was the biggest contributor, accounting for 22.2% of the total OC. Our previous source apportionment results for PM<sub>2.5</sub> in winter in Chengdu also revealed that the contribution of secondary formation was much higher than that of other sources, and that it contributed more than 40% to the total PM<sub>2.5</sub> mass (Zhang et al., 2024a; Zhang et al., 2024e). Therefore, we can conclude that to reduce overall PM<sub>2.5</sub> or OC emissions, greater attention should be focused on atmospheric secondary formation processes. The contribution of vehicular emissions to OC was 17.6%, which is lower than its contribution to PM<sub>2.5</sub>, e.g., 25.6% in winter 2022 in Chengdu (Zhang et al., 2024e). This might be because this study focused on the contribution to OC, whereas when calculating its contribution to PM<sub>2.5</sub>, the secondary products of some gaseous pollutants emitted from vehicular emissions (e.g., nitrate and sulfate from NOx and SO<sub>2</sub>, respectively) would also be included, which would markedly increase the contribution of this source. The contribution of biomass burning was 11.3%, which is much higher than its contribution to PM<sub>2.5</sub> (3.5% in winter 2022) (Zhang et al., 2024e). The low air temperature in winter limited primary biogenic emissions, restricting their contribution to only 5.0%. Despite strict control over dust emissions in the urban area of Chengdu, such as frequent road cleaning and wet removal, in conjunction with continuous spraying at construction sites, dust was the second largest source of OC, accounting for 20.4% of the total OC. This proportion is higher than its contribution to the overall PM<sub>2.5</sub>, e.g., 8.5% in winter 2022 (Zhang et al., 2024e). Currently, in addition to a small amount of regional transmission, local human activities such as traffic and large-scale subway construction might represent the main sources of dust in Chengdu. The plastic related sources was first resolved in Chengdu, contributing 17.4% of the total OC. This OC source not only includes industries that might be related to the emission of plastic OC species, but also incorporates vehicular activities and certain human activities, such as plastic burning, tire wear, and coal combustion. Additionally, the contribution of coal combustion was low, representing only 6.2%, which is much lower than that reported in northern China, e.g., 17% in Zhengzhou (Dong et al., 2024), where coal-fired heating was an important source of OC in winter. Through the above analysis, it is found that specific sources have different importance in relation to reducing PM<sub>2.5</sub> and OC emissions. For example, compared with the overall PM<sub>2.5</sub>, the contributions of dust, biomass burning, and plastic related sources to OC might be worthy of greater attention.

Figure 2: OC source composition based on PMF analysis.


#### 3.4 Evolution of PM<sub>2.5</sub> and organic chemical components and their sources with pollution

To explore the changes in PM<sub>2.5</sub> and organic chemical components and their sources during the pollution evolution process, and to better understand the formation mechanism of heavy pollution, we divided the entire observation period into clean  $(PM_{2.5} 

Figure 3: Chemical composition of (a-c) PM<sub>2.5</sub>, (d-f) OA, and (g-i) OC PMF sources at three pollution levels (C: clean, LP: lightly polluted, HP: heavily polluted).

According to Table S2, the temperature remained at a stable level (6.9–7.7 °C) during the three pollution periods, while the relative humidity showed completely different trends. Although the relative humidity increased by only 2.3% from C to LP period, it increased by 11.1% from LP to HP period. This substantial increase in relative humidity played an important role in promoting the liquid-phase generation of secondary pollutants. Correlation analysis revealed that compared with the weak positive correlation with  $SO_4^{2-}$  ( $R^2 = 0.20$ ), relative humidity exhibited stronger correlation with both  $NO_3^-$  ( $R^2 = 0.47$ ) and NH<sub>4</sub> (R<sup>2</sup> = 0.40), highlighting the important contribution of relative humidity in promoting the generation of secondary inorganic components, especially the latter two, during the process of pollution aggravation. Correspondingly, from C to HP period, the contributions of NO<sub>3</sub><sup>-</sup> and NH<sub>4</sub><sup>+</sup> to PM<sub>2.5</sub> increased by 12.9% and 4.1%, respectively, while that of SO<sub>4</sub><sup>2-</sup> increased by only 0.7% (Fig. 3). Meanwhile, we found that the correlation between  $SO_4^{2-}$  and temperature ( $R^2 = 0.32$ ) was stronger than that between the other two inorganic components ( $NO_3^-$  and  $NH_4^+$ ) and temperature ( $R^2 = 0.13$  and 0.22, respectively). It can be inferred that compared with liquid-phase processes,  $SO_4^{2-}$  was affected more strongly by photochemical processes. However, owing to the minimal change in temperature during the three periods, the contribution of SO<sub>4</sub><sup>2-</sup> also barely changed with the increasing level of pollution. Additionally, according to Table S2, compared with the slight increase in SO<sub>2</sub> (by 0.3 µg m<sup>-3</sup>), the concentration of NO<sub>2</sub> increased by 25.2 µg m<sup>-3</sup> from C to HP period, which provided sufficient precursors for large-scale generation of NO<sub>3</sub><sup>-</sup>, consistent with the largest increase in its contribution. Conversely, the concentration of O<sub>3</sub> declined by 27.5%, which might have led to weakening of photochemical secondary reaction processes and reduction in the associated products. The increase in CO (from 0.6 to 1.1 mg m<sup>-3</sup>) indicates an important contribution from combustion sources, such as biomass burning, in the process of worsening pollution. For the carbonaceous component, although the contribution of EC diminished by 2.2%, the contribution of OC increased by 2.9%. Further comparison revealed that the contribution of primary OC in the total OC decreased from 58.1% to 36.5%, whereas the contribution of SOC increased from 41.9% to 63.5%. Overall, secondary pollutants in PM<sub>2.5</sub> might have been an important cause of heavy pollution formation during our observation period. Furthermore, the NO<sub>3</sub>-/SO<sub>4</sub><sup>2-</sup> ratio increased substantially during the process of worsening pollution (from 2.1 to 3.7), highlighting the important role of mobile sources in the formation of heavy pollution.








For OA compounds, from C to HP period, the contributions of fatty acids and anhydrosugars increased by 6.5% and 3.3%, respectively. Among them, all the increase in the contribution of fatty acids occurred from C to LP period, with an increase of 6.5%, while the contribution remained consistent from LP to HP period. Conversely, the increase in the contribution of anhydrosugars occurred mainly from LP to HP period (by 3.2%), while the contribution increased by only 0.1% from C to LP period. Therefore, fatty acids and anhydrosugars respectively dominated the increase in OA in these two evolutionary processes. Further analysis revealed that, except for C<sub>18:1</sub>, the changes in the contributions of other fatty acids were relatively stable, with a variation of less than 3.5% in the three pollution periods. While the contribution of C<sub>18:1</sub> increased by 19.4% and 2.4% from C to LP and from LP to HP period, respectively, broadly consistent with the evolutionary characteristics of the total

fatty acids. Therefore, we can infer that C<sub>18:1</sub> was the main cause of the changes in the contribution of fatty acids. Previous study found that C<sub>18:1</sub> is derived mainly from vehicular emissions (Rogge et al., 1993), which further highlights the important contribution of this source to the increase in OA in the process of pollution aggravation, consistent with the NO<sub>3</sub><sup>-</sup>/SO<sub>4</sub><sup>2-</sup> ratio analysis results. From LP to HP period, the contribution of levoglucosan in the anhydrosugars diminished by 4.8%, while the contributions of mannosan and galactosan increased by 2.7% and 2.2%, respectively. The corresponding L/M ratio decreased from 5.8 to 4.4. Therefore, the burning of softwood represented an important source for the increased contribution of anhydrosugar species.

From C to HP period, phthalate esters, SOA<sub>S</sub>, and OPAHs were OA species with reduced contributions, with declines of 6.9%, 1.6%, and 1.2%, respectively. Phthalate esters originate mainly from activities such as plastic burning, and their sources were relatively stable. At the same time, some industrial activities related to phthalate emissions would be prohibited during the HP period. Thus, it is reasonable that their contribution experienced the greatest reduction. The SOA<sub>S</sub> and OPAHs were closely related to secondary generation. Therefore, the reduction in their contributions reflected that the changes in sources or environmental conditions during the process of worsening pollution were not conducive to their secondary generation. This is consistent with the observed decrease in O<sub>3</sub> concentration.

For the OC PMF sources, coal combustion and vehicular emissions contributed steadily during the three pollution periods, ranging from 5.9%–6.5% and 16.6%–17.8%, respectively. The contributions of biomass burning and secondary formation increased substantially with a growth rate of 10.7% for them both. Therefore, these two sources were important contributors to the increase in OC during the process of pollution aggravation, consistent with the preceding analysis that revealed increase in the proportion of anhydrosugars (in OA) and SOC (in PM<sub>2.5</sub>) from C to HP period. Conversely, the contributions of primary biogenic emissions, dust, and plastic related sources all showed marked reductions, with decreases of 9.0%, 6.8%, and 6.8%, respectively.

#### 4 Summary and implications

## 4.1 Summary






In this study, PM<sub>2.5</sub> samples collected in Chengdu on a daytime and nighttime basis during a wintertime campaign were characterized for organic molecular compositions. The average concentration of the total organics measured was 2013.4  $\pm$  902.4 ng m<sup>-3</sup>, and that the compounds could be grouped into 15 classes based on their functional groups and sources. The OA were dominated by fatty acids (28.9%), phthalate esters (28.4%), and anhydrosugars (18.0%). The CPI and ratio (such as  $C_{18:0}/C_{16:0}$  and L/M) analysis revealed that anthropogenic sources, such as fossil fuel combustion and biomass burning, were

the main sources of aliphatic lipids. Softwood burning was the main source of anhydrosugars, while the lower temperature during study period had almost no effect on the diurnal variation in the concentrations of sugar compounds. Hopanes and PAHs largely originated from fossil fuel combustion. Although both OPAHs and phthalic acids are associated with the aging of PAHs, their secondary generation mechanisms were notably different. The levels of NOx, relative humidity, and aerosol acidity had substantial impact on the composition and concentrations of  $SOA_I$ , whereas  $SOA_M$  reflected the contribution by  $\alpha$ -pinene. Biomass burning was an important source of three types of biogenic SOA tracers.


Using tracer-based methods, we found that anthropogenic sources, biomass burning, and β-caryophyllene were the three dominant sources of OC, whereas the contributions of other primary (fungal spores and plant debris) or secondary sources (isoprene and α-pinene) to OC were low. The PMF analysis demonstrated that secondary formation (22.2%) was the most important source for OC, followed by dust (20.4%), vehicular emissions (17.6%), plastic related sources (17.4%), biomass burning (11.3%), coal combustion (6.2%), and primary biogenic emissions (5.0%). Meanwhile, the same pollution source had different importance for PM<sub>2.5</sub> and OC. For example, compared with PM<sub>2.5</sub>, the contributions of dust, biomass burning, and plastic related sources to OC might be worthy of greater attention.

Figure 4: Conceptual model of the pollution evolution characteristics and formation mechanisms in this study.

With pollution aggravation, the proportion of secondary inorganic species (especially NO<sub>3</sub><sup>-</sup> and NH<sub>4</sub><sup>+</sup>) and SOC in PM<sub>2.5</sub> increased substantially. Fatty acids and anhydrosugars respectively dominated the increase in OA in these two evolutionary processes. In addition, the PMF results revealed that the increase in OC during the process of worsening pollution was attributable mainly to biomass burning and secondary formation. Our findings provide important information for understanding of the sources, formation mechanisms, and environmental effects of OA in the SCB. Based on the above analysis results, we have created a conceptual model of pollution evolution during the study period (Fig. 4).

## 4.2 Implications



The analysis of the chemical composition and sources of PM<sub>2.5</sub> and OA (or OC), as well as their evolution with pollution in

this study, has provided some new insights. (1) There was substantial difference found in the composition of OC sources between the tracer-based method and the PMF analysis. One of the crucial reasons for this difference is that the tracer-based method is based on only a few representative organic tracers, and the parameters used are based mainly on the measurement results of smog chambers, which might differ greatly from actual atmospheric conditions and ultimately lead to notable uncertainty in the results. Therefore, correct understanding of the source apportionment results obtained by different methods is crucial, which might determine the direction of pollution reduction. Additionally, we must note that large amounts of OC (66.5%) did not appear to be determined and quantified when using the tracer-based method in our study, which could be derived from fossil fuel combustion, amines, dust, proteins, and pollen (Shen et al., 2015; Zhu et al., 2016). (2) For the overall PM<sub>2.5</sub>, we found that secondary inorganic species and SOC were the key species that caused pollution aggravation. Meanwhile, NO<sub>3</sub>-/SO<sub>4</sub><sup>2-</sup> ratio analysis indicated that the contribution of mobile sources increased markedly at the same time. Although mobile sources are often considered typical primary sources, these results are not contradictory. This is because, in addition to emitting primary carbon components such as primary OC and EC, mobile sources also produce NOx, SO<sub>2</sub>, NH<sub>3</sub>, and CO, which are important precursors for secondary inorganic and organic compounds. Currently, Chengdu ranks first among China's cities in terms of the number of motor vehicles. Therefore, vehicular emission control is one of the most important ways to reduce PM<sub>2.5</sub> in Chengdu. Additionally, our previous studies found that regional transmission is an important source of secondary pollutants in Chengdu (Zhang et al., 2024e; Zhang et al., 2024b; Zhang et al., 2025). Therefore, future reduction in PM<sub>2.5</sub> in Chengdu still requires simultaneous control of local emissions and regional transmission. (3) Comparison of the OC PMF results revealed that biomass burning and secondary formation were important sources of increased OC concentration during the process of pollution aggravation. This differs from the evolution of OA molecular composition, which indicated that the contributions of motor vehicles and biomass burning related compounds increased substantially during the process, while the contribution of some SOA compounds diminished. One of the important reasons for this difference is that the GC/MS method can determine only small amounts of OA (7.58% of the total OA mass in this study). Thus, a more comprehensive understanding of OA at the molecular level requires the use of more analytical techniques. Accordingly, it is particularly worthwhile to further explore how to correctly understand the OA measurement results from different perspectives, such as the molecular level and the overall OA, and to integrate different types of results in the future. (4) Some sources that contribute substantially to the OC had very limited contributions to PM<sub>2.5</sub>, and vice versa. This suggests that different measures might be required to reduce PM<sub>2.5</sub> and its typical chemical components, especially when certain chemical components become the focus after the PM<sub>2.5</sub> concentration is reduced in the future.







**Data availability.** The data used in this work can be found at <a href="https://doi.org/10.5281/zenodo.14875327">https://doi.org/10.5281/zenodo.14875327</a> (Junke Zhang et al., 2025).

**Author contributions.** JZ and GW planned this campaign; JZ wrote the paper and led this research; XF and CC performed the data analysis and wrote the manuscript together with JZ; XF, CC, LC, YC and GW conducted experiments and instrument maintenance; YS and SL helped with the data analysis. ASHP made comprehensive revisions and comments on the manuscript.

All authors approved the final version of the manuscript.

Competing interests. The authors declare that they have no conflict of interest.

**Disclaimer.** Publisher's note: Copernicus Publications remains neutral with regard to jurisdictional claims made in the text, published maps, institutional affiliations, or any other geographical representation in this paper. While Copernicus Publications makes every effort to include appropriate place names, the final responsibility lies with the authors.

Financial support. This research has been supported by the National Natural Science Foundation of China (grant nos. U23A2030), the Sichuan Science and Technology Program (grant nos. 2024NSFSC0060), and the Basic Research Cultivation Support Plan of Southwest Jiaotong University (grant no. 2682023ZTPY016).

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
