# Peer review of "Measurement report: Molecular composition, sources, and evolution of atmospheric organic aerosols in a basin city in China"

_EGUsphere, 2025_

## Author Comment (AC1)

Dear reviewer,

Thank you very much for the comments and suggestions, which contribute to improve the quality of our manuscript. We have replied all comments and suggestions in our point-by-point response attached below. In order to highlight the changes what we have done, the color of the revised text will become blue.

In the paper entitled "Measurement Report: Molecular composition, sources, and evolution of atmospheric organic aerosols in a basin city in China", the authors analyzed the atmospheric $PM_{2.5}$ in Chengdu during winter season, especially using GC/MS determined the molecular composition of OA. Firstly, they conducted a analysis of the chemical composition of $PM_{2.5}$ and OA during the observation period; Secondly, they conducted a detailed introduction of the concentration, chemical composition, and sources of OA species. Finally, based on the chemical composition and source apportionment results, they analyzed the characteristics of pollution evolution and proposed the potential implications of this study. As a whole, the logic of the manuscript is reasonable and clear, and a large number of observational results are reported. These pieces of information have reference value for understanding OA in Chengdu and even the Sichuan Basin. However, I have found that there are still some issues in the current manuscript that need to be addressed or clarified, such as Section 3.2 currently contains too many details but appears to be less important information, which needs to be simplified. Therefore, I think a minor revision is necessary before considering acceptance.

1 Introduction. I think the first two paragraphs need to be rewritten. Especially the effects of $PM_{2.5}$ and OA mentioned by the authors seem to be repetitive.

Response. Thanks for this important suggestion. According to the reviewer's comment, we have rewritten these two paragraphs (Line 29-47). The revised content has a clearer logic, which can make it easier for readers to understand the impact of OA and its role in air pollution.

2 Section 2.1. Please provide data sources on the population and motor vehicle ownership in Chengdu city.

Response. These data are sourced from the Chengdu Municipal Statistics Bureau and the Ministry of Ecology and Environment of the People's Republic of China, respectively, and the corresponding website addresses have been added, i.e., "*Chengdu, the capital of Sichuan Province, had a permanent population of 21.4 million in 2023 (https://cdstats.chengdu.gov.cn/). It is also the city in China with the highest number of motor vehicles, i.e., in excess of 6 million vehicles in 2023 (https://www.mee.gov.cn/).*" (Line 93-95).

3 Section 2.3. Reference on "Interagency Monitoring of Protected Visual Environments thermal evolution (IMPROVE) protocol" is necessary.

Response. The corresponding reference has been added, i.e.,

"*Chow, J. C., Watson, J. G., Chen, L. W., Chang, M. C., Robinson, N. F., Trimble, D., and Kohl, S.: The IMPROVE-A temperature protocol for thermal/optical carbon analysis: maintaining consistency with a long-term database, J. Air Waste Manage., 57, 1014-1023, https://doi.org/10.3155/1047-3289.57.9.1014, 2007.*"

4 Line 137. The authors have provided an explanation regarding the data, namely "mean ± standard deviation". Therefore, the "average" before the data is unnecessary. Similar modifications are also required for other parts of the manuscript.

Response. Thanks for this important reminder. We have checked and revised the whole text.

5 Line 148. "OC" in the Figure title should be changed to "OM".

Response. Corrected (Line 148).

6 Line 155. I suggest the author increase the discussion on the contribution of inorganic components, especially in comparison with previous research results.

Response. According to the reviewer's suggestion, we have added a discussion on the contribution of inorganic components, i.e. "*Among the water-soluble inorganic ions, $NO_3^-$ ranked the highest ($25.1 \pm 12.8\ \mu g\ m^{-3}$), followed by $NH_4^+$ ($8.4 \pm 4.2\ \mu g\ m^{-3}$) and $SO_4^{2-}$ ($7.7 \pm 3.0\ \mu g\ m^{-3}$), which accounted for 22.2%, 7.5%, and 7.1% of $PM_{2.5}$ mass, respectively. Compared with the concentrations and contributions of $NO_3^-$ ($15.5 \pm 5.4\ \mu g\ m^{-3}$, 9.8%), $NH_4^+$ ($15.3 \pm 5.7\ \mu g\ m^{-3}$, 9.7%), and $SO_4^{2-}$ ($31.8 \pm 10.7\ \mu g\ m^{-3}$, 20.1%) in winter 2011 in Chengdu (Tao et al., 2014), the concentration and contribution of $NO_3^-$ increased by $9.6\ \mu g\ m^{-3}$ and 12.4%, respectively, while the concentration and contribution of $NH_4^+$ decreased by $6.9\ \mu g\ m^{-3}$ and 2.2%, respectively. Meanwhile, the concentration and contribution of $SO_4^{2-}$ experienced the largest decrease, with a decrease of $24.1\ \mu g\ m^{-3}$ and 13.0%, respectively. This is directly related to the different emission reductions from precursors of these inorganic species (e.g., $SO_2$, NOx, and $NH_3$) in recent years. For example, the annual average $SO_2$ and $NO_2$ concentrations in Chengdu have decreased by 84.2% and 59.3% respectively from 2014 to 2024 (https://sthj.chengdu.gov.cn). The same phenomenon has also been observed in other areas in China, such as Beijing-Tianjin-Hebei (Li et al., 2021a) and Yangtze River Delta (Shen et al., 2020).*" (Line 154-164).

7 Line 224-225. "…between fatty acids and the biomass burning and fossil fuel combustion…". This sentence needs to be expressed more clearly.

Response. This sentence has been revised as "*As validation of the discussion, the correlation analysis found that fatty acids showed strong correlation with the tracers of biomass burning or fossil fuel combustion, e.g., levoglucosan ($R^2 = 0.41$), hopanes ($R^2 = 0.64$), and PAHs ($R^2 = 0.50$).*" (Line 222-224).

8 Line 253-255. "Although levoglucosan dominated the total sugars in our study, its contribution (57.5%) is lower than that reported 10 years ago in urban areas of China (up to 90%) (Wang et al., 2006) " . Is this comparison for the same city? As discussed by the authors, differences in the types of biomass burning in different regions may also lead to variations in the chemical composition of anhydrosugars.

Response. Thanks for this important comment. To the best of our knowledge, there have been no targeted studies on sugar compounds in the urban area of Chengdu in the past. Therefore, we compared our results with another megacity in the SCB, i.e., Chongqing. Due to the fact that Chengdu and Chongqing are both located in the SCB, with similar biomass types and combustion characteristics, the obtained results are comparable. In addition, we also compared the observation results between Chengdu and other representative cities in China, such as Beijing and Guangzhou. This helps us better understand the regional differences in pollution characteristics (Line 261-263).

9 The discussion about "primary saccharides" can be simplified.

Response. According to the reviewer's suggestion, we have simplified this part. Then some unimportant information has been deleted (Line 286-301).

10 Line 308-309. The standard deviation of the average relative humidity should be provided.

Response. The standard deviation of these data has been added, i.e. "*Correspondingly, in this study, the relative humidity at nighttime (85.4 ± 12.6%) was higher than that observed during daytime (71.1 ± 13.1%).*" (Line 306-307).

11 The authors attribute the high concentration of phthalate esters during the daytime to high temperature, however, in other sections they stated that the temperature increase in daytime did not cause an increase in the concentration of OA species, which seems contradictory. I think a reasonable explanation is necessary.

Response. This may be due to differences in the mechanism or degree of influence of temperature on different types of OA species. For example, temperature mainly causes changes in the concentration of phthalate esters by promoting or inhibiting their volatilization process, while the effect of temperature on the concentration of BSOA is mainly through enhancing or weakening its photochemical secondary generation process. In addition, the increase in temperature can also cause the decomposition of some OA components.

12 Line 417. Please provide the full names of "4NC and 4NP".

Response. The full names of "4NC and 4NP" have been added, i.e., "*Despite the differences in concentration levels, similar to previous research (Ren et al., 2024), 4-nitrocatechol (4NC) and 4-nitrophenol (4NP) were found to be the dominant nitrophenol species, contributing 42.4% and 39.2% to the total nitrophenols mass, respectively.*" (Line 406-408). Meanwhile, the full names of all nitrophenols were added in Table S1.

13 Line 534. Is the PMF source analysis result for winter? Generally speaking, the seasonal differences in source composition are significant. Comparing with other seasons can be misleading.

Response. We apologize for the missed season information. Yes, this is a study about winter. Currently, necessary seasonal information has been added, i.e., "*Our previous source apportionment results for PM$_{2.5}$ in winter in Chengdu also revealed that the contribution of secondary formation was much higher than that of other sources, and that it contributed more than 40% to the total PM$_{2.5}$ mass (Zhang et al., 2024a; Zhang et al., 2024e).*" (Line 517-519).

14 Line 595. "relatively stable" is an ambiguous term, and a more accurate expression is necessary.

Response. We have added corresponding data to demonstrate that the changes in the contributions of these species are relatively stable, i.e., "*Further analysis revealed that, except for C$_{18:1}$, the changes in the contributions of other fatty acids were relatively stable, with a variation of less than 3.5% in the three pollution periods.*" (Line 578-579).

15 Regarding the evolution characteristics of pollution. The authors found that during the process of worsening pollution, the contributions of secondary inorganic and SOC both increased significantly. However, the contributions of two types of SOA, namely SOAs and OPAHs, have decreased. What is

the reason for this contradictory phenomenon? Is it due to differences in formation mechanisms?

Response. The secondary inorganic, SOC, or SOA in our study represent the contributions of all secondary inorganic species, secondary organic carbon species, or secondary organic aerosol species, respectively. SOC or SOA contains over thousands of species, with the total concentrations both exceed $10~\mu g~m^{-3}$ during our study period. While SOAs and OPAHs are just two types of substances of SOA, with concentrations of only $70.6 \pm 42.2$ ng $m^{-3}$ and $43.0 \pm 30.0$ ng $m^{-3}$, respectively. Therefore, in the process of increasing pollution, the overall increase in SOA or SOC contribution and the decrease in the contribution of these two substances are not contradictory, as other SOA species may increase.

---

## Author Comment (AC2)

Dear reviewer,

Thank you very much for the comments and suggestions, which contribute to improve the quality of our manuscript. We have replied all comments and suggestions in our point-by-point response attached below. In order to highlight the changes what we have done, the color of the revised text will become blue.

This study gives a very detailed measurement report on molecular composition, sources, and pollution evolution mechanisms of organic aerosols (OA) in $PM_{2.5}$ during winter in a big city Chengdu, Sichuan Basin, China. The methods and analysis are robust with results clearly presented. As the molecular-level OA research for this region is still very limited, this study is helpful understanding the OA formation mechanism in this region and providing insights for air pollution control measures. I recommend the publication of this study after a minor revision.

Minor comments:

1 Add one sentence in the end of the abstract clarifying the significance or implication of this study.

Response. According to the reviewer's suggestion, the necessary sentence has been added, i.e., "*These results are of great value for understanding the characteristics and formation mechanisms of OA, and its contribution to air pollution in the SCB.*" (Line 25-27).

2 The use of GC/MS to quantify fatty acids, phthalate esters, and anhydrosugars etc, is well-established. However, for some more volatile organic compounds (e.g., isoprene SOA tracers), could the authors clarify potential losses of these relatively high volatility compounds during derivatization and describe correction measures?

Response. Thank you very much for this very important comment. Yes, we completely agree with that part of $SOA_I$ tracers like 3-MeTHF-3,4-diols, C5-alkene triols, and 2-methyltetrols would evaporate during thermal desorption and/or derivatization heating processes. Howere, the losses caused by this decomposition are difficult to determine (Lopez-Hilfiker et al., 2016). Therefore, as with previous study (Li et al., 2018), we have added necessary explanations in the section S1 of the supplementary materials, i.e., "*In addition, it is worth nothing that part of detected $SOA_I$ tracers like 3-MeTHF-3,4-diols, C5-alkene triols, and 2-methyltetrols would evaporate during thermal desorption and/or derivatization heating processes. Thus these tracers in the current study were possibly somewhat underestimated.*" (Line 34-37). This can enable readers to have a more accurate understanding of our study results. Meanwhile, the same measurement method has also been widely applied in the previous studies (Li et al., 2013; Cui et al., 2023; Fu et al., 2011; Fu et al., 2012; Wan et al., 2019). Therefore, we believe that with necessary explanations, this volatilization will not affect the credibility of our results.

References

Cui, L. L., Gao, Y., Chen, Y. B., Li, R., Bing, H. J., Wu, Y. H., and Wang, G. H.: Chemical characteristics and source apportionment of biogenic primary and secondary organic aerosols in an alpine ecosystem of Tibetan Plateau, J. Geophys. Res.-Atmos., 128, e2022JD037897, https://doi.org/10.1029/2022jd037897, 2023.

Fu, P. Q., Kawamura, K., and Miura, K.: Molecular characterization of marine organic aerosols collected during a round-the-world cruise, J. Geophys. Res.-Atmos., 116, D13302,

https://doi.org/10.1029/2011jd015604, 2011.

Fu, P. Q., Kawamura, K., Chen, J., Li, J., Sun, Y. L., Liu, Y., Tachibana, E., Aggarwal, S. G., Okuzawa, K., Tanimoto, H., Kanaya, Y., and Wang, Z. F.: Diurnal variations of organic molecular tracers and stable carbon isotopic composition in atmospheric aerosols over Mt. Tai in the North China Plain: an influence of biomass burning, Atmos. Chem. Phys., 12, 8359-8375, https://doi.org/10.5194/acp-12-8359-2012, 2012.

Li, J. J., Wang, G. H., Cao, J. J., Wang, X. M., and Zhang, R. J.: Observation of biogenic secondary organic aerosols in the atmosphere of a mountain site in central China: temperature and relative humidity effects, Atmos. Chem. Phys., 13, 11535-11549, https://doi.org/10.5194/acp-13-11535-2013, 2013.

Li, J. J., Wang, G. H., Wu, C., Cao, C., Ren, Y. Q., Wang, J. Y., Li, J., Cao, J. J., Zeng, L. M., and Zhu, T.: Characterization of isoprene-derived secondary organic aerosols at a rural site in North China Plain with implications for anthropogenic pollution effects, Sci. Rep., 8, 535, https://doi.org/10.1038/s41598-017-18983-7, 2018.

Lopez-Hilfiker, F. D., Mohr, C., D'Ambro, E. L., Lutz, A., Riedel, T. P., Gaston, C. J., Iyer, S., Zhang, Z., Gold, A., Surratt, J. D., Lee, B. H., Kurten, T., Hu, W. W., Jimenez, J., Hallquist, M., and Thornton, J. A.: Molecular composition and volatility of organic aerosol in the Southeastern U.S.: implications for IEPDX derived SOA, Environ. Sci. Technol., 50, 2200-2209, https://doi.org/10.1021/acs.est.5b04769, 2016.

Wan, X., Kang, S. C., Rupakheti, M., Zhang, Q. G., Tripathee, L., Guo, J. M., Chen, P. F., Rupakheti, D., Panday, A. K., Lawrence, M. G., Kawamura, K., and Cong, Z. Y.: Molecular characterization of organic aerosols in the Kathmandu Valley, Nepal: insights into primary and secondary sources, Atmos. Chem. Phys., 19, 2725-2747, https://doi.org/10.5194/acp-19-2725-2019, 2019.

3 For the PMF model, the factor of plastic-related sources is interesting. Could the authors include specific markers, e.g., styrene derivatives or tire-wear tracers, to strengthen the credibility of this factor?

Response. Thank you very much for this important suggestion. We fully agree with the reviewer's comment that additional tracers are beneficial in strengthening the credibility of this factor. Unfortunately, this study did not measure these species. However, we believe that the current PMF results are still reliable. This is because: (1) The measurement method and results of phthalate esters in this study are reliable. Meanwhile, a large number of previous studies have indicated that phthalate esters are used as plasticizers in synthetic polymers and a softener in polyvinylchloride, and they have been used as the important tracers for plastic emission over the Arctic Ocean (Fu et al., 2013), in Mt. Tai (Fu et al., 2012), in Kathmandu Valley (Wan et al., 2019), and in fourteen Chinese cities (Wang et al., 2006). Correspondingly, in our PMF analysis results, these species appeared extensively in the source profiles of plastic-related sources. (2) Similar to our study, in some previous studies, such as Zhu et al. (2022), Liu et al. (2024), Kang et al. (2018), and Gadi et al. (2019), they also identified plastic-related sources mainly based on phthalate esters in PMF without measuring other tracers. Thanks again for this important comment, which have important guiding value for our future study.

References

Fu, P. Q., Kawamura, K., Chen, J., Charrière, B., and Sempéré, R.: Organic molecular composition of marine aerosols over the Arctic Ocean in summer: contributions of primary emission and secondary aerosol formation, Biogeosciences, 10, 653-667, https://doi.org/10.5194/bg-10-653-2013, 2013.

Fu, P. Q., Kawamura, K., Chen, J., Li, J., Sun, Y. L., Liu, Y., Tachibana, E., Aggarwal, S. G., Okuzawa, K., Tanimoto, H., Kanaya, Y., and Wang, Z. F.: Diurnal variations of organic molecular tracers and stable carbon isotopic composition in atmospheric aerosols over Mt. Tai in the North China Plain: an influence

of biomass burning, Atmos. Chem. Phys., 12, 8359-8375, https://doi.org/10.5194/acp-12-8359-2012, 2012.

Gadi, R., Shivani, Sharma, S. K., and Mandal, T. K.: Source apportionment and health risk assessment of organic constituents in fine ambient aerosols (PM$_{2.5}$): a complete year study over National Capital Region of India, Chemosphere, 221, 583-596, https://doi.org/10.1016/j.chemosphere.2019.01.067, 2019.

Kang, M. J., Fu, P. Q., Kawamura, K., Yang, F., Zhang, H. L., Zang, Z. C., Ren, H., Ren, L. J., Zhao, Y., Sun, Y. L., and Wang, Z. F.: Characterization of biogenic primary and secondary organic aerosols in the marine atmosphere over the East China Sea, Atmos. Chem. Phys., 18, 13947-13967, https://doi.org/10.5194/acp-18-13947-2018, 2018.

Liu, Y. L., Shen, M. X., Liu, H. J., Dai, W. T., Qi, W. N., Zhang, Y. F., Li, L., Cao, Y., Wang, X., Guo, X., Jiang, Y. K., and Li, J. J.: Molecular compositions and sources of organic aerosols at a rural site on the Guanzhong Plain, Northwest China: the importance of biomass burning, Particuology, 89, 44-56, https://doi.org/10.1016/j.partic.2023.10.014, 2024.

Wan, X., Kang, S. C., Rupakheti, M., Zhang, Q. G., Tripathee, L., Guo, J. M., Chen, P. F., Rupakheti, D., Panday, A. K., Lawrence, M. G., Kawamura, K., and Cong, Z. Y.: Molecular characterization of organic aerosols in the Kathmandu Valley, Nepal: insights into primary and secondary sources, Atmos. Chem. Phys., 19, 2725-2747, https://doi.org/10.5194/acp-19-2725-2019, 2019.

Wang, G. H., Kawamura, K., Lee, S., Ho, K. F., and Cao, J. J.: Molecular, seasonal, and spatial distributions of organic aerosols from fourteen Chinese cities, Environ. Sci. Technol., 40, 4619-4625, https://doi.org/10.1021/es060291x, 2006.

Zhu, Y. H., Tilgner, A., Hans Hoffmann, E., Herrmann, H., Kawamura, K., Xue, L. K.., Yang, L. X., and Wang, W. X.: Molecular distributions of dicarboxylic acids, oxocarboxylic acids, and α-dicarbonyls in aerosols over Tuoji Island in the Bohai Sea: effects of East Asian continental outflow, Atmos. Res., 272, 106154, https://doi.org/10.1016/j.atmosres.2022.106154, 2022.

4 Add references for Lines 357-360 about the formation mechanisms of OPAHs and its role in formation of ROS.

Response. Thank you very much for this important reminder, and the necessary references have been added, i.e.,

*"Cassee, F. R., Heroux, M. E., Gerlofs-Nijland, M. E., and Kelly, F. J.: Particulate matter beyond mass: recent health evidence on the role of fractions, chemical constituents and sources of emission, Inhal. Toxicol., 25, 802-812, https://doi.org/10.3109/08958378.2013.850127, 2013.*

*Chung, M. Y., Lazaro, R. A., Lim, D., Jackson, J., Lyon, J., Rendulic, D., and Hasson, A. S.: Aerosol-borne quinones and reactive oxygen species generation by particulate matter extracts, Environ. Sci. Technol., 40, 4880-4886, https://doi.org/10.1021/es0515957, 2006.*

*Shin, S. M., Lee, J. Y., Shin, H. J., and Kim, Y. P.: Seasonal variation and source apportionment of oxygenated polycyclic aromatic hydrocarbons (OPAHs) and polycyclic aromatic hydrocarbons (PAHs) in PM$_{2.5}$ in Seoul, Korea, Atmos. Environ., 272, 118937, https://doi.org/10.1016/j.atmosenv.2022.118937, 2022."*